# The Influence of New Surveillance Data on Predictive Species Distribution Modeling of *Aedes aegypti* and *Aedes albopictus* in the United States

**DOI:** 10.3390/insects10110400

**Published:** 2019-11-12

**Authors:** Hannah S. Tiffin, Steven T. Peper, Alexander N. Wilson-Fallon, Katelyn M. Haydett, Guofeng Cao, Steven M. Presley

**Affiliations:** 1Department of Environmental Toxicology, Texas Tech University, Lubbock, TX, 79416, USA; hsg14@psu.edu (H.S.T.); steve.peper@ttu.edu (S.T.P.); afallon65@gmail.com (A.N.W.-F.); haydettk@msu.edu (K.M.H.); 2Department of Geography, Texas Tech University, Lubbock, TX, 79409, USA

**Keywords:** *Aedes aegypti*, *Aedes albopictus*, MaxEnt, species distribution modeling, mosquitoes

## Abstract

The recent emergence or reemergence of various vector-borne diseases makes the knowledge of disease vectors’ presence and distribution of paramount concern for protecting national human and animal health. While several studies have modeled *Aedes aegypti* or *Aedes albopictus* distributions in the past five years, studies at a large scale can miss the complexities that contribute to a species’ distribution. Many localities in the United States have lacked or had sporadic surveillance conducted for these two species. To address these gaps in the current knowledge of *Ae. aegypti* and *Ae. albopictus* distributions in the United States, surveillance was focused on areas in Texas at the margins of their known ranges and in localities that had little or no surveillance conducted in the past. This information was used with a global database of occurrence records to create a predictive model of these two species’ distributions in the United States. Additionally, the surveillance data from Texas was used to determine the influence of new data from the margins of a species’ known range on predicted species’ suitability maps. This information is critical in determining where to focus resources for the future and continued surveillance for these two species of medical concern.

## 1. Introduction

With the recent emergence and reemergence of numerous vector-borne diseases, knowledge of disease vectors’ presence and distribution has become of paramount concern for protecting human and animal health, as well as protecting national economies and national defense. *Aedes (Stegomyia) aegypti* (Linnaeus) (Diptera: Culicidae) and *Ae. (Stegomyia) albopictus* (Skuse) (Diptera: Culicidae) have posed a very real and consistent threat to human health in the tropical and sub-tropical regions of the world, and have recently gained greater attention in the United States as well. These species are known to transmit numerous pathogens of critical health importance, including chikungunya, dengue, yellow fever and Zika viruses [1,2,3,4]. The emergence and rapid spread of Zika virus, coupled with its devastating effects upon infants, and the establishment of the local transmission of Zika, chikungunya and dengue viruses in the United States in recent years, has renewed interest in the distribution and biology of these vectors [2,3].

Several studies have modeled one or both of these species’ distributions in the United States in the past five years [2,3,5], as well as modeled distributions on a global scale [4,6,7]. However, studies at such a large scale, nationally and especially globally, can miss the many complexities that contribute to a species’ distribution and establishment within microhabitats. Additionally, many localities in the United States, and globally, have only had sporadic surveillance for these two species, or have never had surveillance conducted. To address these common and well-known difficulties in determining the known and predicted suitable ranges for *Ae. aegypti* and *Ae. albopictus,* this study focused on collecting additional surveillance data on the edge of the known range of these species and in other areas that have lacked surveillance in Texas, United States, a state with several incidents of local transmission of viruses transmitted by these two species. Data was collected at a finer scale than the county-level, and was used along with a data repository compiled by Kraemer and colleagues [1] of all records of *Ae. aegypti* and *Ae. albopictus* collected during the years of 1960–2014 to model their distributions in the United States.

The first objective of this study was to build a model with high predictive performance for these two species, as well as to elucidate the primary bioclimatic variables responsible for their current and predicted distributions. A second objective was to evaluate the effects of surveillance at the edges of current known species’ ranges on model output, as determining habitat suitability at the edges of their current known range is imperative in planning for impending further range expansion [5]. These models can be used as a tool to promote surveillance and control in key locations of medium to high habitat suitability to aid the future planning and management of these vectors to limit disease and disease outbreaks caused by these two species.

## 2. Materials and Methods

### 2.1. Field Data Collection and Processing 2016–2017

Surveillance for gravid females of both species was conducted using ovitraps during August–November 2016 in 28 counties of the Texas Panhandle region [8], and during June–October 2017 in 33 counties throughout Texas [9]. These months were chosen based on typical peaks in the activity and abundance of mosquitoes in this region. Surveillance kits were delivered to agencies and individuals that agreed to aid this research initiative. Surveillance kits included 473 mL black plastic cups and seed germination paper to line the inside of these cups to serve as the ovitraps for mosquitoes. Kits also included binder clips to clip the germination paper to the cups, datasheets and instructions for using the ovitraps, Ziploc™ bags and pre-paid United States Postal Service (USPS) envelopes to mail the used germination strips back to the Vector-Borne Zoonoses Laboratory (VZL) at Texas Tech University. Collaborators were asked to fill ovitraps half-way with water and place the ovitraps in five locations that were at least 25% shaded throughout their jurisdiction for 4–5 days. Ovitraps were placed in areas with higher human density, such as parks, schools and residential homes to increase the likelihood of a gravid female of one of these two species laying eggs on the seed germination paper in the ovitraps. One ovitrap was placed at each location, with five ovitraps deployed per county.

Collaborators submitted potentially egg-laden germination papers via USPS to the VZL at Texas Tech University for rearing to adulthood and speciation. A total of 704 germination paper strips were processed in 2016 from 28 different counties in the Texas Panhandle Region and a total of 561 germination strips were processed in 2017 from 33 counties throughout Texas. As these species were being tested for insecticide resistance as part of another project, the five germination strips were pooled together into one emergence chamber to ensure sufficient numbers of mosquitoes were reared consistent with established protocols for testing insecticide resistance. To utilize location data effectively for this study, a central location between the five collection locations was calculated and used as the presence/absence location in the model. The data was visually inspected for repetitive and erroneous location data. Repeated locations were excluded from the database, with any record for presence given priority to remain in the database.

### 2.2. Additional Ae. aegypti and Ae. albopictus Data

Additional data was used to construct the species distribution models (SDMs) from an open-access online database publicly available by Kraemer and colleagues [1] with all known occurrence records of *Ae. aegypti* and *Ae. albopictus* globally between the years of 1960–2014. This database is the most comprehensive dataset compiled of all known occurrence locations for *Ae. aegypti* and *Ae. albopictus* [4,10]. When these locations were precisely indicated, they were coded as “point” locations at 5 km^2^ spatial resolution in this database. When records were not precise locations, for example large cities or approximate locations that could not be assigned a location value at the 5 km^2^ resolution, they were termed “polygons”, and they were classified according to their polygon resolution (5–10 km^2^, 10–25 km^2^, 25–100 km^2^, or greater than 100 km^2^). Data collected by the VZL team was geocoded using Texas A and M’s batch geocoding service, and organized and coded for consistency with the global Kraemer and colleagues [10] dataset.

### 2.3. Bioclimatic Variables

Meteorological variables are known to influence *Ae. aegypti* and *Ae. albopictus* growth, reproduction, and survivability, with temperature and precipitation being particularly important variables towards predicting habitat suitability for these species [4,10,11,12]. Temperature and precipitation bioclimatic variables averaged from 1970–2000 at a resolution of 2.5 min were obtained from the WorldClim database version 2, with this resolution used, as it most closely approximates the spatial resolution of the mosquito occurrence records database [13]. WorldClim provides freely accessible, high resolution, global climate datasets at different resolutions, compiled with data collected from weather stations located all over the world.

The bioclimatic variables used in this study were chosen according to a literature review of variables previously employed in other species distribution models, with a focus upon models that used MaxEnt, and modeled the distributions of one or both of these species [1,3,4,10,12,14]. As the primary objective was to build a species distribution model with a high predictive performance, the covariates employed were those that were most directly related to the biology and reproductive capabilities of these two species.

Frequently, bioclimatic variables are highly correlated, which can overinflate estimates of the model’s predictive ability, as well as mask the contribution of one covariate behind the contribution of another when the two are closely related [3,15]. To avoid this issue of multi-collinearity, a Pearson’s correlation matrix was produced in R Studio version 1.1.4 [16] to exclude highly correlated variables from being used in the model. Any variables with a correlation greater than 0.80 were excluded from the model [3]. Multiple iterations of the model were run with one covariate of the highly correlated pair included to determine which covariate had the greatest permutation importance, and thus contributed to the best predictive model.

### 2.4. Species Distribution Modeling Using MaxEnt in R Studio

A first set of SDM models was developed using all USA data (VZL surveillance data and data available in the online database). To examine the influence on model output of data collected from areas with limited surveillance records and located in areas previously predicted not to have high suitability for these species, the surveillance data collected by the VZL team in 2016 and 2017 in Texas was excluded from the dataset to create a second set of SDMs. All other model parameters were kept constant between the two sets of models, including background records and covariates, for a direct comparison of the VZL data’s influence without additional confounding variables in model development and output.

MaxEnt is a machine-learning software designed for presence-only species distribution modeling, and has been shown to outperform other frequently used SDM applications, particularly when used with presence-only data and/or with small sample sizes [17,18,19]. MaxEnt outputs several statistics to aid in the consideration of covariates employed in the model, as well as in assessing overall model strength, accuracy and performance.

Of these, the AUC value (area under the curve of the receiver operating characteristic curve—ROC) was used as the overall measure of model strength with the goal of obtaining the highest AUC value possible with a combination of the bioclimatic variables imported from WorldClim. An AUC value of 0.5 indicates that the model is no better than a random prediction, while an AUC value of 1.0 indicates a perfect prediction [3,17,18]. It should be noted that the highest achievable AUC value using presence-only data is below 1.0 [18].

### 2.5. Background Data

Using R Studio, background points were randomly generated for use in the model, with presence points excluded from being chosen as background points, and background points corrected for random sampling, being weighted by the size of the cells in the coordinate system [20,21]. A ratio of background points to occurrence records was used to determine the number of background points used for model generation [3,4]. A 1:1 presence to the background ratio was used for all SDMs of *Ae. aegypti* and a 2:1 presence to background ratio was employed for the SDM of *Ae. albopictus* using all occurrence records in the United States. At first a 1:1 ratio was used for the SDM of *Ae. albopictus*, but this resulted in a very conservative and unrealistic prediction of its distribution. Since the goal of the SDM is to promote continued and additional surveillance, the more liberal SDM was desired, and consequently, the 2:1 ratio was used due to the abundance of occurrence records for *Ae. albopictus* in the United States.

### 2.6. Covariate Selection Process

All bioclimatic variables downloaded from WorldClim’s database with a Pearson’s correlation coefficient below 0.80 were included in the original models. Generally, variables with a permutation coefficient below 5% were excluded one by one from the model, with different combinations of these variables employed until the highest AUC value possible was achieved [3]. A ten-fold cross validation was used to determine the model’s predictive strength [3,4].

Permutation importance and variable response curves were used as the main determinants for variables’ inclusion and removal, as these are known to show variable importance and contribution to the model [3,18]. Permutation importance is an especially important statistic in determining the variables to include and exclude from the final model [3]. The variables used in the final models were those that resulted in the highest mean AUC values.

## 3. Results

### 3.1. Model 1. All USA Data

The first set of models (Figure 1) contained a total of 480 *Ae. aegypti* occurrence records (Model 1a) and 1671 *Ae. albopictus* occurrence records (Model 1b). Both models predict high habitat suitability across the southern United States. The SDM predicts a much broader range northward for *Ae. albopictus* than is predicted for *Ae. aegypti*. While both models show low habitat suitability in the Rocky Mountain Range in the west, low habitat suitability in the Appalachian Mountain Range in the east is particularly noticeable in the SDM for *Ae. albopictus,* with a narrow band of low suitability extending along the range from Pennsylvania into Georgia. This area of low habitat suitability is present in the SDM of *Ae. aegypti* as well, but it is less apparent, as habitat suitability is predicted to be low this far north on either side of the Appalachians.

The final predictive model for the distribution of *Ae. aegypti* in the United States (Model 1a) had a mean AUC value of 0.944. This final model included the following bioclimatic parameters: The maximum temperature of the warmest month (Bio5), the minimum temperature of the coldest month (Bio6) and the precipitation of the wettest month (Bio13) (Table 1). The variable that had the greatest contribution to this model was the minimum temperature of the coldest month (Bio6, permutation importance = 88.4%). While the contribution statistics for Bio13 were much lower than that of Bio6, its permutation importance was still well above the 5% threshold for exclusion from the model and consequently contributed to the final model’s predictive performance (PI = 9.3%).

While Bio5 had a permutation importance below the 5% threshold for exclusion, it was shown to be inherently important in the final model as when excluded the model’s mean training AUC value consistently decreased (Bio5 included: mean AUC = 0.944; Bio5 excluded: mean AUC = 0.920).

The response curves for these variables indicate that the highest predicted habitat suitability for *Ae. aegypti* is in habitats where temperatures peak at approximately 10 °C for the minimum temperature during the coldest month of the year. Habitat suitability peaked at approximately 70 mm of accumulated precipitation during the wettest month, with a sharp decrease in suitability below 70 mm, and a relatively gradual decrease in suitability above 70 mm (Figure 2).

The final predictive model of *Ae. albopictus* distribution in the United States (Model 1b) had a mean AUC value of 0.883. There were four bioclimatic variables that proved essential in the development of this final model: Mean diurnal range (Bio2), maximum temperature of the warmest month (Bio5), temperature annual range (Bio7) and precipitation of the wettest month (Bio13) (Table 2). Two of these variables were the same as those included in the final predictive model of *Ae. aegypti* distribution, Bio5 and Bio13. However, both of these variables had greater permutation importance in the SDM of *Ae. albopictus* (Bio5 PI = 27.2%; Bio13 PI = 26.6%). Additionally, three of the four variables had similar reported contributions to the model when considering both the variables’ percent contribution and their permutation importance (PC range = 24.7; PI range = 15.1), indicating that the interplay of these factors may be highly important to the environmental suitability for *Ae. albopictus* as opposed to the primary importance of one variable. While Bio2 was below the permutation importance threshold of 5% for exclusion from the final model, the mean AUC value was consistently lower when Bio2 was excluded (Bio2 included: Mean AUC = 0.883; Bio2 excluded: Mean AUC = 0.879).

The response curves of these variables show that the highest predicted habitat suitability for *Ae. albopictus* is in habitats with temperatures above 30 °C, and annually receive more than 80 mm precipitation, and without extreme fluctuations in mean diurnal temperatures (<15° range of maximum–minimum mean monthly temperatures) (Figure 3).

### 3.2. Model 2. VZL Data Excluded

A second set of species distribution models (Figure 4) contained a total of 444 *Ae. aegypti* occurrence records (Model 2a) and 1600 *Ae. albopictus* occurrence records (Model 2b). These models were constructed excluding surveillance data collected by the VZL team during 2016 and 2017, and they used the same approximate ratios of presence to background points that were used in the construction of Model 1 (*Ae. aegypti*: *n* = 444, background = 480; *Ae. albopictus*: *n* = 1600, background = 835). These two models were constructed with the goal of determining the influence of precise surveillance data collected on the margins of the current known distribution and in areas with little to no surveillance conducted in the past on SDM output.

The species distribution model of *Ae. aegypti* (Model 2a) with the VZL data excluded resulted in a mean AUC value of 0.935. This model resulted in high suitability for *Ae. aegypti* predicted across most of the southern states in the United States and in western and central California (suitability > 0.8). However, a substantial portion of the United States at a latitude above 35° North was predicted to have low suitability for this species (suitability ≤ 0.2). The western Panhandle Region of Texas was also predicted to have very low suitability for this species (suitability ≤ 0.2).

The variable with the highest permutation importance remained the same as in Model 1a, with the minimum temperature of the coldest month (Bio6) resulting in a much higher permutation importance than either of the other two covariates (Bio6 PI = 92.0%, Table 3). The other two covariates had very similar permutation importance values, with a difference of only 1.1% between the two covariates.

The response curves of Bio5 and Bio13 show abrupt changes in the predicted value in response to the independent variable (Figure 5). The response curve of Bio5 shows particularly abrupt changes in this predicted value at two temperatures in particular. At approximately 30 °C, there is a sudden but gradual decrease in the predicted value, and an abrupt and steep increase in the predicted value at 40 °C.

The species distribution model of *Ae. albopictus* (Model 2b) with VZL data excluded resulted in a mean AUC value of 0.847. This SDM predicted high suitability for *Ae. albopictus* east of central Texas and throughout a substantial portion of the Midwest and eastern United States (suitability > 0.8). However, very low suitability was predicted along the Appalachian Mountain range in the eastern United States (suitability ≤ 0.2). Low suitability was also predicted in most of the Northeast and in the northern Midwest, approximately above a latitude of 40° N (suitability ≤ 0.2), except for southern Michigan, where the SDM predicted medium suitability (suitability = 0.4). Additionally, low suitability was predicted west of central Texas (suitability ≤ 0.2), except for in a few locations in California, where medium to high suitability was predicted (suitability ≥ 0.4).

Temperature annual range (Bio7) had the highest value for the permutation importance of the covariates used in this SDM (PI = 33.9%, Table 4). Permutation importance values had a lower range between all of the covariates in this model compared to Model 1b (Model 2b range = 16.7%; Model 1b range = 37.1%). Additionally, permutation importance values for all of the covariates met the threshold of 5% permutation importance for inclusion in the model. The covariate with the lowest permutation importance value was the mean diurnal range (Bio2), with a much higher value compared to its value in Model 1b (Model 2b PI = 17.2%; Model 1b PI = 4.6%).

The response curves for Model 2b remain similar to those of Model 1b. The main difference between the models is that the response curves had smoother and less abrupt changes in the predicted values in Model 1b compared to those in Model 2b (Figure 6). However, the relationships between the predicted values and the independent variables remained very similar between the two models.

## 4. Discussion

### 4.1. Response Curves

The response curves of the variables included in these models could be used to determine suitable habitat characteristics, and consequently, locations of surveillance priority. The most important variables for predicting habitat suitability differed between the two mosquito species. For *Ae. Aegypti*, the most important variable according to the model was the minimum temperature during the coldest month of the year (Bio6). This is unsurprising, given its constrained poleward expansion compared to the known occurrence records of *Ae. Albopictus*, which has been found in cooler locations.

Additionally, precipitation was an important determinant in *Ae. aegypti* habitat suitability with a peak at approximately 70 mm during the wettest month, and a sharp decrease in suitability below this amount of precipitation during the wettest month of the year. This could be used as a threshold to demarcate areas for new or enhanced surveillance strategies and approaches.

While the response curve of the minimum temperature of the coldest month for Models 1a and 2a of *Ae. aegypti* remain consistent between the two models (Figure 2b and Figure 5b), the two other response curves differ between the two models (Figure 2a,c and Figure 5a,c). The response curve of the maximum temperature of the warmest month shows greater variability in predicted suitability when VZL data is excluded (Figure 2a and Figure 5a). This may be due to the low number of records of *Ae. aegypti* in the United States, thus when occurrence records are excluded and the available records to build the model are decreased, the model results in greater variability due to this low sample size. The greatest observed difference in response curves between the models is between Figure 2c and Figure 5c, which show the precipitation during the wettest month. The response curve for Model 1a (Figure 2c) has a steeper slope in decreasing suitability after suitability reaches its peak at approximately 70 mm of precipitation, compared to a more gradual decline in suitability as precipitation increases after reaching peak suitability at approximately 120 mm of precipitation in Model 2a (Figure 5c). This may be due to the inclusion of occurrence records from arid areas in the West Texas Panhandle region in Model 1a. These areas receive low amounts of rainfall and were predicted to be unsuitable for these species of mosquitoes in previous SDMs, however these were found in several counties in this arid region during surveillance in 2016 and 2017. This may have contributed to the change in the response curve when these more arid locations were excluded from the model, with the response curve reflecting the lower importance of rainfall after a certain amount of precipitation has occurred. Numerous studies have shown the decreased susceptibility of *Ae. aegypti* eggs to desiccation as well as longer egg survival times and lower egg mortality independent of habitat conditions, compared to increased mortality to *Ae. albopictus* eggs during dry periods and egg survivability being highly dependent on relative humidity and temperature [22,23,24]. Additionally, these results highlight the importance of microclimates. While overall the Panhandle Region of Texas is semi-arid and rural, microclimates must exist with enough precipitation and human or animal presence to enable these mosquitoes to survive in these habitats, possibly due to availability of human containers.

The response curves of the variables used in the development of the SDM for *Ae. albopictus* show the highest predicted habitat suitability for *Ae. albopictus* is in an environment that is warm (>30 °C during the warmest month), receives ample rainfall (>80 mm during the wettest month), and without extreme fluctuations in mean diurnal temperatures (<15° range of maximum–minimum mean monthly temperatures) (Figure 3). This aligns with the known biology of this species and the characteristics of its tropical and sub-tropical home range. The variable that has the greatest influence on predicted habitat suitability is that of temperature annual range (Bio7). This is surprising given that several studies show that *Ae. albopictus* can exist at a broader temperature range, particularly regarding cooler temperatures, compared to *Ae. aegypti* [1,2,23], and that instead moisture requirements tend to be of greater importance to this species compared to *Ae. aegypti* [22,23,24]. However, as the response curves indicate, Bio7 only results in decreased habitat suitability at approximately 38 °C, which is a relatively extreme annual temperature range. This range would be a good way to quickly and easily demarcate areas as uninhabitable for this species, as its predicted suitability in an environment with >40 °C range in annual temperatures rapidly decreases with zero predicted suitability at a range of approximately 50 °C.

### 4.2. Evaluation and Comparison of SDMs

The models created in this study had high AUC values and finer detail in the predicted habitat suitabilities than many of the previously created models [3,4,14]. This is likely due to the use of specific occurrence data locations in both the global database and with the addition of VZL data from areas in the western Texas Panhandle Region that had only limited data available before the surveillance effort in 2016.

While Ding and colleagues [4] also used this same global database, they implemented it on a global scale and merged occurrence records that were within a specified proximity to one another, thus compromising precision. The habitat suitability’s predicted by the SDMs created in this study were similar to those predicted by the SDMs created by Johnson and colleagues [3]. As stated earlier, comparatively the models in this study resulted in finer detail, likely due to Johnson and others [3] using occurrence records at the county-level as well as dichotomizing their results based on different thresholds for habitat suitability. Additionally, Model 1b predicted portions of the western Texas Panhandle Region to have medium to high suitability for *Ae. albopictus*, where these previous models predicted low suitability [3,4]. This represents another item of note- the importance of new surveillance and the influence that additional data can have on a species distribution model.

Another major difference in the predicted suitability of the models in this study compared to previous studies was the low suitability predicted along the Appalachian Mountain Range in the eastern United States. This area of low suitability is present in both SDMs, but is much more pronounced in the SDM of *Ae. albopictus* as it has high suitability predicted on either side of the mountain range. To the authors’ knowledge, this distinctive zone of low suitability has not been reported in other SDMs of these species. However, a study by Watts and colleagues [25] reports that elevation can serve as a fairly accurate representative for the probability of occurrence of *Ae. aegypti* and its historical transmission of dengue in Brazil, particularly above elevation thresholds of 1600 and 2000 m. The highest elevations within the Appalachian Mountain range are within the area spanning from northern Georgia to southern Pennsylvania, with the highest peak reaching over 2000 m, and peaks in the entire range averaging 900 m. While elevation was not used as a covariate in these models, its residual effects on habitat suitability were still evident in the models. This could be due to the strong influence of minimum temperature of the coldest month on *Ae. aegypti*’s predicted habitat suitability and the strong influence of temperature annual range for *Ae. albopictus*. Both of these covariates likely have a high correlation with elevation, and so this relationship may be what is depicted in the models. This type of detail in habitat suitability has not been predicted in previous species distribution models for these species, and consequently could be used as an effective and efficient tool for public health and vector control agencies for determining new and additional locations to conduct surveillance for these critical disease vectoring species.

### 4.3. Data along Margins of Current Known Distributions

The influence of precise data from new locations, especially in locations near the margins of the current known or current predicted extent of these species, was investigated in the development of Model 2 when VZL data was excluded from the models’ input. Both Models 2a and 2b (Figure 4) resulted in a constricted prediction of habitat suitability compared to the inclusion of VZL data in Models 1a and 1b (Figure 1). The influence of the precise VZL data was more clearly seen with the SDM of *Ae. albopictus* (Model 2b), as more occurrence records were collected for this species by the VZL and nationwide in the global database, compared to *Ae. aegypti*.

As can be observed when comparing SDMs of the same species (Model 1a compared to Model 2a and Model 1b compared to Model 2b), lower habitat suitability was predicted when the VZL data were excluded from the model. While the same states are predicted to be suitable for this species between Model 1b and Model 2b, there are more areas of medium suitability predicted at the margins of the predicted species range (suitability = 0.4). Additionally, there is a substantial change in predicted habitat suitability in Texas, with very low suitability predicted in most of the western Texas Panhandle Region when the VZL data was excluded (suitability ≤ 0.2). This trend suggests the importance of inclusion of new and precise surveillance data into model development as it contributes to more accurate SDMs and can aid in public health vector control efforts. Without the VZL data, the Panhandle region of Texas was predicted to have very low suitability for this species. However, after conducting surveillance in 2016 and 2017 it was determined that this species is present in many counties throughout this region.

Additionally, surveillance focused along these margins of the current known distribution can aid in determining suitability in other locations along the fringe of the current known distribution. For example, when the VZL data was excluded in the SDM of *Ae. albopictus,* the predicted suitability for this species was not only diminished in Texas but in other locations as well, particularly in southern and central California, where new occurrences of *Ae. aegypti* and *Ae. albopictus* have been reported [26]. This further exemplifies the importance of new and precise surveillance, especially along these edges of the current known and predicted distributions of these two medically important species. It should be noted that these SDMs are not to be interpreted as known ranges for these two species, but rather used as tools for determining areas of medium to high habitat suitability. This information can be used to dedicate surveillance and vector control resources to areas of medium to high habitat suitability where these species are not yet known to be present but have the habitat conditions likely conducive to their survival if introduced. While *Ae. aegypti* and *Ae. albopictus* do not travel far distances on their own, they can and have been transported great distances through human trade and travel, with the tire trade particularly implicated for the movement of these species [1,3]. These SDMs provide another valuable, and inexpensive tool with the era of open-access datasets and software to determine suitable habitat to proactively survey for these species for increased knowledge and capacity to manage future potential disease risk and introduction. Additionally, these models highlight the importance of active surveillance methods at the margins of current predicted habitat distributions to continue to improve these species distribution models, further refining the use and applications of SDMs as a tool in vector surveillance and control.

### 4.4. Potential Limitations

Because entomological field surveys are conducted to determine the presence of these vectors for public health reasons, there is sampling bias in the location placement of ovitraps. Ovitraps are often placed in areas of medium or high human activity to monitor for the presence of these vectors in areas that would be more at risk by their presence and potential to transmit pathogens of human health concern. In future studies, subsampling these areas could be employed to determine the effects and effective application of this method. However, since these SDMs were being created with the final goal of proactively promoting and supporting surveillance in additional locations throughout the United States, a more liberal SDM was desired and subsampling could have resulted in a more conservative prediction, as well as lowered the models’ predictive power. Another potential limitation of this study is the timeframes used in the development of these SDMs. While the timeframe used for the bioclimatic variables overlapped with the years of mosquito occurrence records used, it did not encapsulate the full timeframe of occurrence records as the most recent dataset available from WorldClim was from 1970–2000. This dataset was used as it provided the bioclimatic variables desired for these models while still providing a thirty-year span of data. Future model iterations may benefit from using more current datasets of bioclimatic variables, especially if weather events continue to become more severe and less predictable compared to recorded historic conditions. Lastly, there are always limitations with developing species distribution models and these models should be used as predictors of suitability to focus and aid surveillance efforts, not as defined species distributions.

## 5. Conclusions

With the increasing availability of large-scale, widely accessible datasets, as well as improvements in species distribution modeling techniques such as the relatively recent introduction of machine learning methods like MaxEnt and boosted regression trees (BRT), modeling distributions has become more common [17,18,27,28]. SDMs have been utilized as decision-making tools in a wide variety of disciplines, from conservation planning, to land management, to disease and vector risk mapping [16]. This type of technology has also recently gained considerable attention in the infectious and vector-borne disease community [27,29]. In recent years, a considerable number of species distribution models for either *Ae. aegypti, Ae. albopictus*, or both have been created [1,4,7]. With the exception of the SDMs created by Johnson and colleagues [3], all of the aforementioned SDMs were created on a global scale. The SDMs created in this study sought to address habitat suitability on a national scale in the United States, as well as to address the influence of additional surveillance data on model output and predictive strength.

For the purposes of this study, a liberal, informative estimate of habitat suitability for these vectors was desired to promote surveillance in areas that either do not conduct surveillance frequently or have never conducted surveillance for these two critical species. The models developed that excluded the precise data collected by the VZL team in the western Texas Panhandle Region in 2016 and throughout Texas in 2017 further highlighted the importance of additional surveillance in areas previously lacking surveillance, especially in locations near the margins of the current known extent of these species. Promoting increased surveillance is critical in establishing a preemptive defense against the diseases that these two species can transmit. The recent surge of dengue fever and yellow fever in the Americas, as well as the recent emergence and rapid dispersal of Zika virus throughout the Americas, including the United States, makes it clear that increased surveillance is a critical first line of defense against the introduction and establishment of these viruses in the United States.

## Figures and Tables

**Figure 1 insects-10-00400-f001:**
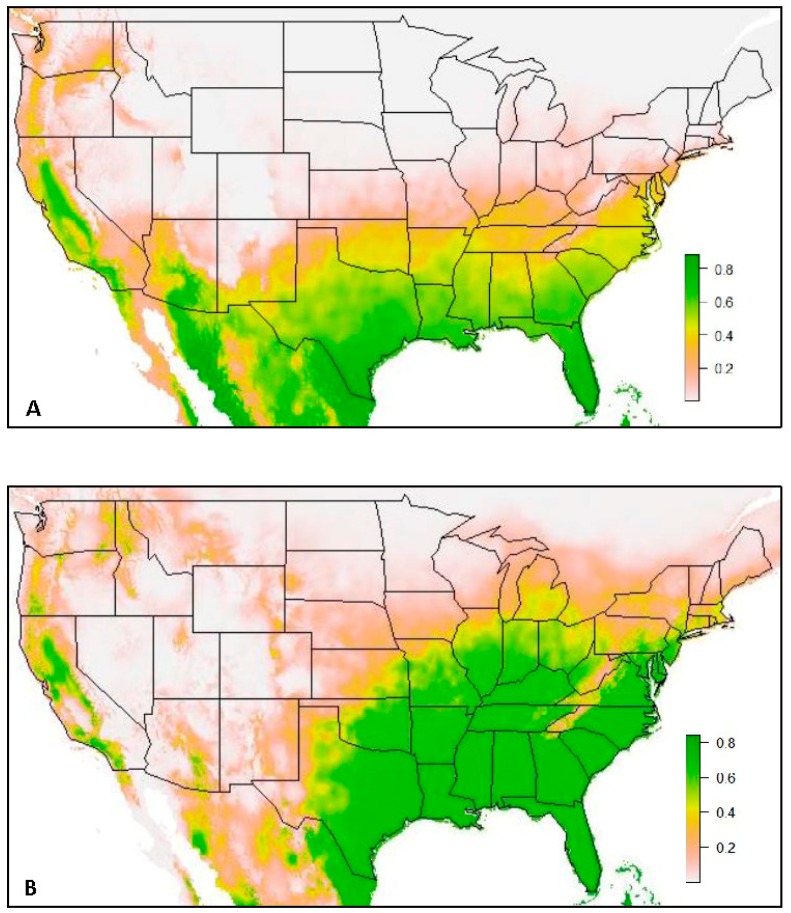
(**A**) Model 1a: Species distribution model of *Ae. aegypti*, and (**B**) Model 1b: Species distribution model of *Ae. albopictus*, both using all occurrence records from 1960–2014 (Kraemer et al. 2015b database) and records obtained by VZL personnel in Texas during 2016 and 2017. Green coloration indicates areas of high predicted habitat suitability (suitability ≥ 0.6), with darker green indicating areas with highest habitat suitability (suitability ≥ 0.8), yellow coloration indicates moderate habitat suitability (suitability = 0.4–0.6), tan coloration indicates low habitat suitability (suitability = 0.2–0.4) and white to light tan coloration indicates extremely low habitat suitability (suitability = 0.0–0.2).

**Figure 2 insects-10-00400-f002:**
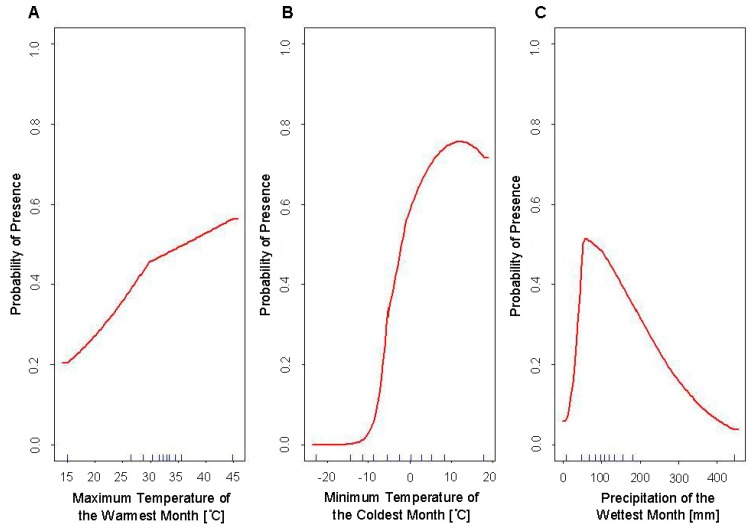
Response curves of the variables used in the final SDMs of *Ae. aegypti* (Model 1a) using all occurrence records showing predicted habitat suitability to (**A**) the maximum temperature of the warmest month (Bio5), (**B**) the minimum temperature of the coldest month (Bio6), and (**C**) the precipitation of the wettest month (Bio13).

**Figure 3 insects-10-00400-f003:**
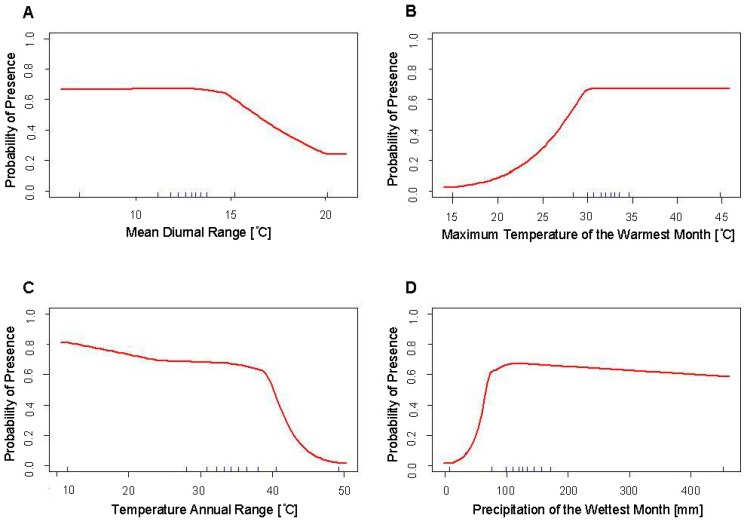
Response curves of the variables used in the final SDMs of *Ae. albopictus* (Model 1b) using all occurrence records showing predicted habitat suitability to (**A**) mean diurnal range (Bio2), (**B**) maximum temperature of the warmest month (Bio5), (**C**) temperature annual range (Bio7), and (**D**) precipitation of the wettest month (Bio13).

**Figure 4 insects-10-00400-f004:**
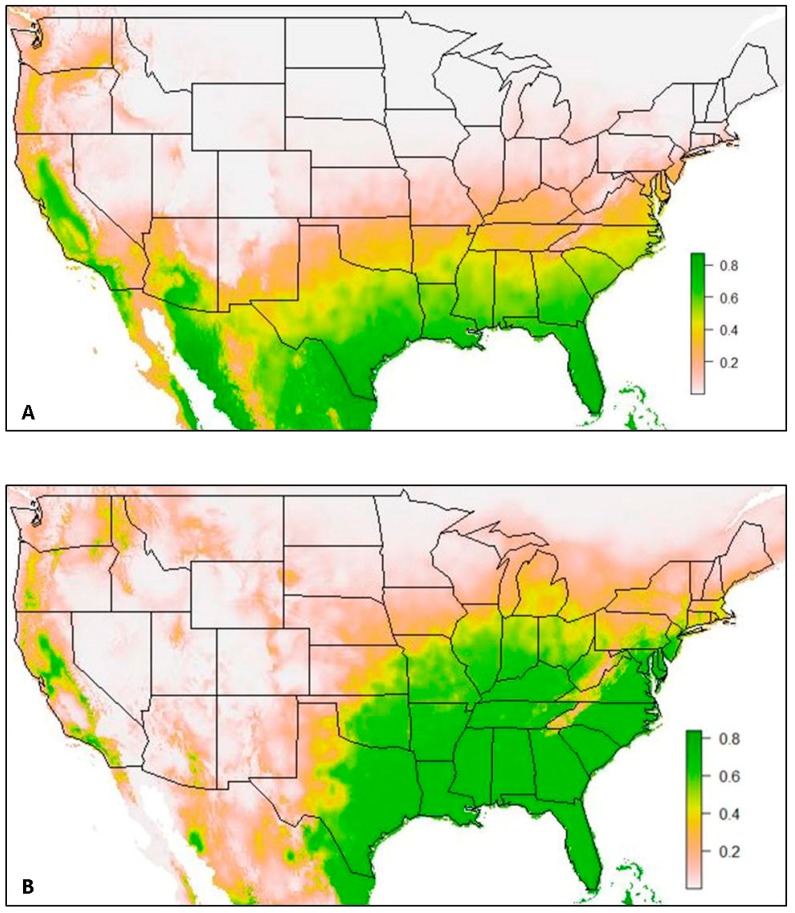
(**A**) Model 2a: Species distribution model of *Ae. aegypti*, and (**B**) Model 2b: Species distribution model of *Ae. albopictus*, both using all occurrence records from 1960–2014 (Kraemer et al. 2015b database) and excluding the 2016 and 2017 VZL data from model input. Green coloration indicates areas of high predicted habitat suitability (suitability ≥ 0.6), with darker green indicating areas with highest habitat suitability (suitability ≥ 0.8), yellow coloration indicates moderate habitat suitability (suitability = 0.4–0.6), tan coloration indicates low habitat suitability (suitability = 0.2–0.4) and white to light tan coloration indicates extremely low habitat suitability (suitability = 0.0–0.2).

**Figure 5 insects-10-00400-f005:**
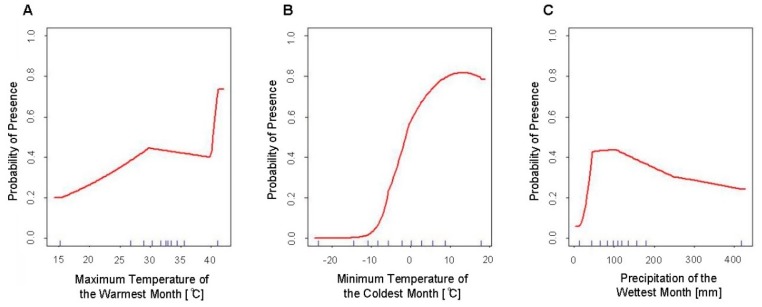
Response curves of the variables used in the final SDMs of the *Ae. aegypti* (Model 2a) using occurrence records excluding the VZL data, showing predicted habitat suitability to (**A**) the maximum temperature of the warmest month (Bio5), (**B**) the minimum temperature of the coldest month (Bio6) and (**C**) the precipitation of the wettest month (Bio13).

**Figure 6 insects-10-00400-f006:**
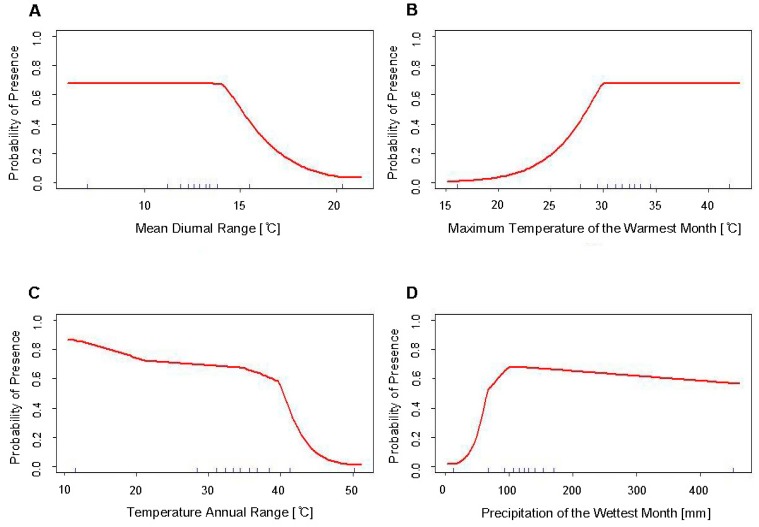
Response curves of the variables used in the final SDMs of *Ae. albopictus* (Model 2b) using occurrence records excluding VZL data, showing predicted habitat suitability to (**A**) mean diurnal range (Bio2), (**B**) maximum temperature of the warmest month (Bio5), (**C**) temperature annual range (Bio7) and (**D**) precipitation of the wettest month (Bio13).

**Table 1 insects-10-00400-t001:** Model 1a covariates included in the species distribution model (SDM) of *Ae. aegypti*, using all occurrence records of *Ae. aegypti* distribution in the United States, in order of highest to lowest permutation importance.

Variable	Percent Contribution	Permutation Importance
Min. Temperature of Coldest Month (Bio6)	92.1	88.4
Precipitation of Wettest Month (Bio13)	5.1	9.3
Max. Temperature of Warmest Month (Bio5)	2.7	2.3

**Table 2 insects-10-00400-t002:** Model 1b covariates included in SDM of *Ae. albopictus,* using all occurrence records of *Ae. albopictus* distribution in the United States, in order of highest to lowest permutation importance.

Variable	Percent Contribution	Permutation Importance
Temperature Annual Range (Bio7)	39.4	41.7
Max. Temperature of Warmest Month (Bio5)	16	27.2
Precipitation of Wettest Month (Bio13)	40.7	26.6
Mean Diurnal Range (Bio2)	3.8	4.6

**Table 3 insects-10-00400-t003:** Model 2a covariates included in SDM of *Ae. aegypti* excluding VZL data, in order of highest to lowest permutation importance.

Variable	Percent Contribution	Permutation Importance
Min. Temperature of Coldest Month (Bio6)	92.6	92.0
Precipitation of Wettest Month (Bio13)	2.7	4.5
Max. Temperature of Warmest Month (Bio5)	4.7	3.4

**Table 4 insects-10-00400-t004:** Model 2b covariates included in the SDM of *Ae. albopictus* excluding VZL data, in order of highest to lowest permutation importance.

Variable	Percent Contribution	Permutation Importance
Temperature Annual Range (Bio7)	31.8	33.9
Max. Temperature of Warmest Month (Bio5)	16.7	29.4
Precipitation of Wettest Month (Bio13)	43.1	19.6
Mean Diurnal Range (Bio2)	8.5	17.2

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
