# Peer review of "The Influence of New Surveillance Data on Predictive Species Distribution Modeling of *Aedes aegypti* and *Aedes albopictus* in the United States"

_insects, 2019, doi:10.3390/insects10110400_

Round 1

Reviewer 1 Report

The present manuscript describes the development of a predictive model for Aedes aegypti and Aedes albopictus mosquitoes distribution, by the integration of updated field collection data and global database of occurrence records. The authors were able to refine the predictive distributions for Texas and USA. Moreover, they were able to demonstrate how new surveyed data can impact the prediction and distribution models. 

The article is well-written and brings a clear message about the importance of constant surveillance to estimate and report more precise distribution dynamics of these vectors. I consider these article accepted after minor revision: 

1-Even though the authors have cited the article containing the information about the Texas collections, it would be nice to add a phrase reporting that the detailed information about the traps and methods of collections are found in [10]. The description is not complete in the present paper. 

2-The figure legends would include a brief description of the meaning of the colors. Despite the index bar exhibited inside of each figure, the readers will welcome a simplified description (for instance, "green - high probability", etc)

3-The resolution of the figures 2 and 3 is not the best, at least on my file. The authors could improve it for a better final version. 

4- I would suggest a modification that would affect a little bit the structure of the present manuscript, but it would improve readers readability and comparative analysis. I would suggest figures 1 and 2 side-by-side, with 4 panels. This way, readers can easily compare the outcome of the 2 different models. Perhaps, a third figure could be added to demonstrate this side-by-side comparison, even though the maps will be smaller than in figure 1 and 2. 

Reviewer 2 Report

It is a great initiative, and the idea of using models and open data banks sounds the best way to share information about these species, correlating with different parameters.
I do have concerns with the data collection, especially for those performed on the field. The methods section doesn't mention the total amount of ovitraps deployed, but it does indicate the reference #9. Which says there were five ovitraps used in each locality, and that they were collected weekly, 28 out of 32 areas were evaluated, while in methods section says only 5, which were the criteria and parameters for the shortlist. I also believe that the chosen period is related to winter or seasonality (dry/rainy season) - However, a more continuous collection period would be recommended to support the findings.
Also, there is a disconnection over the different data type. Field collection was done in 2016-2017, while the additional global distribution data goes from 1960-2014, and more intriguing was the bioclimatic data that comprises 1970-2000. My point is that with all issues regarding global warming and climatic changes, almost 20 years of climatic data is missing, which I believe can contribute to the prediction power of the model.
Another second point is that it is unclear the number presented as the occurrence records for Ae. aegypti and Ae. albopictus. The difference between the values on model 1 and 2 (for each species - A and B) is tiny and due to the variations of data collection period is not surprising that just a small amount of data provided such a big difference, especially for Ae. aegypti which the model presents to be more related to "min. temperature of coldest month (Bio06)". And again, how the data of some areas can influence the entire country with a great diversity of biomas as the south of the USA?
I understand the trouble collecting field data to evaluate and to increase accuracy for the model, and the objective of the study makes it clear. But I don't think the lack of overlapping the data collection period can be ignored, and I would like to see its impact on the global model.
As a predictive tool, I think the model could include a human cultural component that could be linked with bioclimatic parameters and social-environmental condition.

Reviewer 3 Report

The manuscript “The influence of new surveillance data on predictive species distribution modelling of Aedes aegypti and Aedes albopictus in the United States” by Tiffin et al. contains interesting insights on the effects of the inclusion of relatively modest datasets on the outputs of predictive models, notably in different sections of the country. Anything that can help prioritise surveillance for these invasive mosquito species is potentially useful, particularly as temperatures are predicted to rise in response to climate change and we are seeing emerging infectious diseases, such as Zika and chikungunya viruses, in unprecedented numbers.

The results were for the most part in strong agreement with the known biology of Aedes aegypti and Ae. albopictus. In general, this paper could do with further interpretation of the results (see questions below for specific examples), both in terms of why particular variables were or were not influential in the model, and also what we should think about predictive models overall based on these results.

SPECIFIC COMMENTS:

INTRODUCTION

Lines 32-34. “…to human health in the tropics and sub-tropical regions of the world, they have recently gained greater attention in the United States as well”. This is odd phrasing, considering that the southern United States is part of the subtropical world.

Line 34. At the risk of being pedantic, “vector” is not used as a verb in the sense of transmitting a pathogen. Use “transmit” instead.

MATERIALS AND METHODS

Lines 65-74. It’s not clear how many VZL traps were placed and where. This might be in reference #10, but I found it locked behind a paywall. It would be good to summarise it here. How many traps per county?  I assume one trap per premises? Any detail about the finer distribution would be good as well, if available (e.g. how far apart were the traps placed within an county?).

RESULTS

Figure 2C. It is not surprising that probability of presence for Ae. aegypti would increase with increasing precipitation. A bit of discussion on the decrease of probability of presence with increasing precipitation

DISCUSSION

267-311 Under “Response Curves” subsection, the text here is a summary of the results. This would be more appropriate under Results, whereas the Discussion should go into further detail or interpretation.

267-269. “The response curves of the variables included…could be used to determine…locations of surveillance priority”. I think the discussion is missing the idea that invasive mosquitoes generally have to come from somewhere else, i.e. invasion pathways. We know that Ae. aegypti and Ae. albopictus don’t fly very far, so much of their movement happens when they are transported along with human movement (in used tires, etc.). These “invasion pathways” are also factored in when determining surveillance priority. While the models in this paper predict the probability of finding Ae. aegypti and albopictus based on environmental condition that lead to their survival, it is important to also consider the likelihood that mosquitoes are going to show up there in the first place. I realize that invasion pathways are beyond the scope of this study, but I think it would be useful to identify as a limitation in the Discussion.

298-311. For example, the models for Aedes albopictus suggest a higher dependence on rainfall. One could draw parallels from previous works such as Reiskind and Lounibus 2012 Medical and Veterinary Entomology 27(4): 421-429, which finds that Aedes aegypti tends to outcompete Ae. albopictus in drier parts of Florida. Or Kraemer et al 2015 (reference #1 in this manuscript), which also found Ae. albopictus to have a greater rainfall dependence (postulating that Aedes aegypti may thrive in drier areas because domestic containers are refilled as part of gardening, etc., whereas Ae. albopictus in peridomestic sites may be more reliant on rain to fill their breeding sites).

355-357. “As can be observed when comparing SDMs of the same species…, lower habitat suitability was predicted when the VZL data was excluded from the model”.   I guess this is the main result of the paper. The paper also demonstrates that adding data adds to the resolution of the model. What is not as clear is, does this mean that predictive models are generally not useful due to being based on an insufficient number of data points? Should we trust predictive models? How do we know when a model is based on sufficient data to be trusted? Discussion on these questions would be very interesting.

Reviewer 4 Report

The manuscript "The influence of new surveillance data on predictive species distribution modeling of Aedes aegypti and Aedes albopictus in the United States" presents an interesting analysis about habitat suitability for the yellow fever mosquito and the Asian Tiger mosquito in the contiguous 48 states of USA. 

Overall the analysis seems to have been well done. A few comments for minor clarifications are included:

1) How was the background data for models 2a and 2b generated? Please, explain in detail (see line 209).

2) The importance of the manuscript will be even more clear if a brief note about the population living in the area with the two vectors is included, specifically. how many more people are likely to be affected by these vectors when the data from the Texas Panhandle is included or excluded?.

3) It will be also nice to see an exact figure, measured with area units, about the difference in area occupied by both species under model 1 and model 2.
